# Comparison of Properties of PVA Nanocomposites Containing Reduced Graphene Oxide and Functionalized Graphene

**DOI:** 10.3390/polym11030450

**Published:** 2019-03-08

**Authors:** Gi Tae Park, Jin-Hae Chang

**Affiliations:** Department of Polymer Science and Engineering, Kumoh National Institute of Technology, Gumi 39177, Korea; pkgt0129@naver.com

**Keywords:** poly(vinyl alcohol) (PVA), reduced graphene oxide, functionalized-graphene sheet, nanocomposite

## Abstract

The thermal properties, morphologies, oxygen barrier properties, and electrical conductivities of poly(vinyl alcohol) (PVA) hybrid films containing different nanofillers were compared. For the fabrication of the PVA hybrid films, we used reduced graphene oxide (RGO) synthesized from graphite or functionalized hexadecylamine-graphene sheets (HDA-GS) obtained from HDA and GS as a reinforcing filler. The properties of the PVA hybrid films fabricated by intercalating PVA and the fillers for different filler contents ranging from 3 to 10% *w*/*w* were then compared. The dispersions of the graphene fillers in the matrix polymers were examined using wide-angle X-ray diffraction and field emission scanning electron microscopy, and the changes in their thermal properties were observed using differential scanning calorimetry and thermogravimetric analysis. Moreover, we measured the oxygen permeability and electrical conductivity of the films to investigate their industrial applications. In addition, all the physical properties of the PVA composites obtained using the two nanofillers were compared.

## 1. Introduction

Poly(vinyl alcohol) (PVA) is a colorless and transparent hydrophilic polymer that has been widely used in industry for a long time [1,2,3]. PVA is known to have strong oxygen barrier properties due to the strong hydrogen bonding of the hydroxy (OH) groups in PVA, as well as better tensile strength, elongation, and abrasion resistance than general synthetic resins. PVA is also currently used in various fields such as fibers, films, adhesives, surface treatment agents, and coating materials [4,5,6]. However, due to the hydrophilicity of PVA, its mechanical properties and electrical properties vary considerably depending on the external humidity [7,8,9], and its solubility, viscosity, and film strength may also differ depending on the degree of saponification (DS) [4,10].

A number of PVA nanocomposites have been reported using various well-known nanofillers, particularly clay [11,12,13]. When the clay layer is dispersed and mixed well into PVA gel, even in the wet or partially dry state, it forms a nanocomposite that is hybridized at the nanometer scale [14,15,16]. However, the clay layers agglomerate at the microscale level when they are heated under vacuum during the drying process. Therefore, the production of PVA/clay nanocomposites is difficult from a practical viewpoint [17].

On the other hand, graphene, which has been investigated extensively as a nanofiller, is an aggregate containing numerous benzene rings and has very good thermal and mechanical properties, high electron mobility, and very large surface area. Graphene also shows good optical transparency and has the potential for application in transparent electrodes and solar cells [18,19,20]. However, it is difficult to obtain large quantities of graphene with current manufacturing methods, and it is also difficult to utilize the characteristics of pure graphene [21,22]. Also, graphene should have good dispersion and compatibility with matrix polymers to achieve polymeric nanocomposites that are used in the form of two-dimensional carbon products. Since widely used pure carbon materials are difficult to melt or dissolve, chemical or physical modification is required to increase their dispersibility and compatibility with matrix polymers for use in polymer nanocomposites [23,24,25,26]. In order to achieve the dispersibility required in such polymer nanocomposites, graphene must first be functionalized into functionalized-graphene sheets (F-GS) with various organic moieties to increase its organophilicity [27,28,29].

In general, F-GS has its advantages to be an ideal nano-sized filler for polymer nanocomposites: firstly, the F-GS still possess most physical properties of pure graphene even if there are some changes in chemical structure; secondly, the chemical modifying on the surface of GS can enhance the dispersion of GS in the polymer matrix, and also the interfacial interaction between GS and the polymer. In order to improve the dispersibility of the polymer hybrid, a hexadecylamine-graphene sheet containing some oxygen (HDA-GS) was synthesized by a reaction with hexadecylamine (HDA), which resulted in the removal of most of the epoxy groups via deoxygenation [30]. 

Graphite-based materials are commercial polymer fillers used in polymer melt compounding to improve polymer properties such as thermo-mechanical properties, abrasion resistance, flame resistance, thermo-electrical conductivities, electromagnetic shielding, and gas barrier properties. For various potential applications, it would be desirable to exfoliate GS and to disperse the individual GS within the polymer matrix. Since GSs are electrically conducting and optically transparent, they can be applied to produce ultrathin films useful as transparent devices and electrodes. 

In our lab, reduced graphene oxide (RGO) has been obtained by chemical reaction treatment using carbon as the base material. F-GS functionalized with various structures, including various alkyl and phenyl groups, have been synthesized using graphene oxide [31]. Figure 1 shows the chemical structures of various F-GS that were synthesized for polymer nanocomposites. In addition, the so-called position-selective F-GS substituted in both the horizontal and vertical directions have also been synthesized [32]. 

In this study, PVA hybrid films were prepared using two graphene-based synthetic nanofillers, namely, RGO and HDA-GS which were synthesized by a chemical method. The thermal properties, morphology, oxygen permeability, and electrical conductivity of PVA hybrids containing two nanofillers were investigated. Further, the physical properties of PVA hybrids were also compared.

## 2. Materials and Methods

### 2.1. Materials

Carbon flakes with a mesh size of 75 mesh were used in this study (Sigma Aldrich, Tokyo, Japan). Organic solvents were used without further purification. PVA was obtained from SKC (Seoul, Korea) and had a degree of polymerization of 1700 and a degree of saponification of 85 to 88% mol/mol. GO, RGO, and HDA-GS were directly synthesized in the laboratory; *N,N*’-dimethylacetamide (DMAc) and hydrazine hydrate (NH_2_NH_2_·H_2_O) were purchased from Sigma Aldrich Chemical Co. (Seoul, Korea).

### 2.2. Preparation of GO

GO was prepared on the basis of the well-known methods of Hummers and Stankovich [33,34]. The simple synthetic method was as follows: First, sulfuric acid (H_2_SO_4_) and 1 g of graphite were mixed in an ice bath at 0 °C. After the solution was mixed and dispersed, 2 g of sodium acetate (CH_3_COONa) was added, and the solution was stirred for 10 min. 10 g of potassium permanganate (KMnO_4_) was then dissolved in the sulfuric acid solution for 10 min and dispersed at 30 °C for 12 h. The reaction was terminated by adding 20 mL of hydrogen peroxide (H_2_O_2_) to the solution. The solution was then diluted by being poured into 2 L of distilled water to remove KMnO_4_. This solution was washed several times with distilled water, neutralized to pH = 6–7, and freeze-dried to obtain GO.

### 2.3. Synthesis of Reduced GO

1 g of GO was dispersed in 1 L of distilled water, and 10 mL of hydrazine hydrate was added to the GO solution. The resulting mixture was vigorously stirred at 80 °C for 1 h. The mixture was washed 2–3 times with distilled water/ethanol (1:1 *v*/*v*) and then dried at 80 °C in a vacuum oven [35]. An overview of the synthetic method is shown in Scheme 1.

### 2.4. Synthesis of HDA-GS

Graphene oxide (GO) was synthesized from natural graphite using a multistep route known as Hummers’ method [33]. HDA-GS was then synthesized from HDA and GO using the following procedure: GO (1 g) was dissolved in 1.5 L of distilled water. HDA (2.00 g; 8.28 × 10^−3^ mol) was added to 25 mL of ethanol; this mixture was stirred at 25 °C under a steady stream of N_2_ and subsequently added to the GO/water mixture. The resulting mixture was heated for 12 h at 25 °C under a steady stream of N_2_, washed twice with a mixture of distilled water and ethanol (1:1 *v*/*v*), and dried under vacuum at 70 °C for 24 h to obtain HDA-GS. The synthesized HDA-GS still contains oxygen elements. The synthetic route for HDA-GS is shown in Scheme 1.

### 2.5. Synthesis of PVA Hybrids with HDA-GS

Nanocomposites were prepared using the solution intercalation method. Since the PVA hybrids were synthesized using either RGO or HDA-GS, and the amount and composition of the two fillers were almost the same, only the method using HDA-GS is described in this paper. Although the PVA/RGO hybrid used only water as the solvent, the subsequent synthesis of the RGO–PVA hybrid followed the same method as the HDA-GS hybrid.

HDA-GS was dispersed in DMAc for 5 h, and the pre-dispersed HDA-GS was then added to the prepared PVA solutions to achieve HDA-GS contents of 3, 5, 7, and 10% *w*/*w*. For example, using 5% *w*/*w* PVA/HDA-GS as an example, 0.0263 g of HDA-GS was added to 2.35 g (2.5 mL) of DMAc and dispersed for 5 h by stirring with ultrasonication. 3.90 g (DMAc 3.62 mL, 3.4 g) of the prepared PVA solution was added to the HDA-GS solution, yielding a PVA/HDA-GS mixed solution with a solid content of 8%. After 5 h of stirring at room temperature, the air bubbles formed during the reaction were removed, the solution was poured into a glass plate prepared in advance, and the solvent was removed in a 40 °C oven over 72 h to obtain a film.

### 2.6. Characterizations

Fourier-transform infrared (FT-IR) spectra were obtained via an FT-IR 460 (JASCO, Tokyo, Japan) instrument in the range 4000–600 cm^−1^ using KBr pellets. Solid-state Nuclear Magnetic Resonance (NMR) experiments were performed using a Bruker 400 NMR spectrometer. ^13^C cross-polarization magic angle spinning (CP/MAS) NMR experiments were performed at a Larmor frequency of 100.58 MHz. The powdered sample was placed in a 7 mm CP/MAS probe. The magic angle spinning rate was 3 kHz to minimize spinning sideband overlap.

Differential scanning calorimetry (DSC) (SINCO, S-650, Tokyo, Japan) and thermogravimetric analysis (TGA) (TA analyzer, Q 500, New Castle, DE, USA) were conducted to measure the thermal properties of the obtained PVA hybrid film. Measurements were conducted using heating and cooling rates of 20 °C/min under a nitrogen atmosphere.

A wide-angle X-ray diffractometry (WAXD) was performed to determine the variation in interlayer distance between GO, RGO, and the HDA-GS hybrids obtained from raw carbon. In this case, a Rigaku (D/Max-IIIB, Tokyo, Japan) diffractometer equipped with a Ni-filter was used with a Cu-Kα target. Measurements were conducted from 2° to 32° at a rate of 2 °/min at room temperature. Field emission scanning electron microscopy (FE-SEM) (JEOL, JSM-6500F, Tokyo, Japan) was conducted to confirm the morphology of the hybrid. An SPI sputter coater was used to sputter the fractured surfaces with gold to enhance their conductivity.

A gas permeation tester (MOCON DL 100, OX-TRAN 2/61, Minneapolis, MN, USA) was used to measure the oxygen transmission rate (O_2_TR). The measurement was carried out in accordance with American society for testing and materials (ASTM) D3985, and the conditions were set to an O_2_ concentration of 100% and a relative humidity (RH) of 0%. The O_2_TRs of the films were determined at 23 °C and at a pressure of 760 Torr. The electrical conductivity of the hybrid film was measured using the four-point probe of a surface resistance meter (AIT, CMT-SR1000N, Seoul, Korea). The thickness of the obtained film was measured using a digital micrometer (Mituyoto 293-240, Tokyo, Japan).

## 3. Results and Discussion

### 3.1. FT-IR and ^13^C-NMR Spectroscopy

Figure 2 shows the FT-IR spectra of PVA, graphite, GO, RGO, and HDA-GS. The spectrum of pure graphite does not exhibit any peaks, while that of PVA and GO exhibit significant and broad absorption peaks characteristic of the OH and COOH functional groups [31]. The details of each spectrum are described below.

The characteristic absorption peaks of PVA were observed at 3330 cm^−1^ (O–H stretching), 2940 cm^−1^ and 2910 cm^−1^ (asymmetric CH_2_ stretching), 1730 cm^−1^ (water absorption), and 1256 and 1090 cm^−1^ (C–H bending and C–O stretching). In the case of the GO, the characteristic absorption peak of the O–H was observed at 3220 cm^−1^ (stretching). The asymmetric epoxy peak appeared at 3063 cm^−1^ but did not appear to overlap with the OH peak. Characteristic peaks were also observed at 1732 and 1616 cm^−1^ (C=O stretching) and 1040 cm^−1^ (C–O stretching). The epoxide ring and C=C bond peaks were considerably weaker compared to the others, whereas the O–H stretching peak was more intense.

In the RGO spectrum, all the peaks were significantly smaller or had disappeared (Figure 2). This indicated that the oxygen moieties in RGO were almost, but not completely, eliminated. The ^13^C-NMR data showed similar trends to the FT-IR spectra. As the oxygen moieties in GO (Figure 3) were removed, the aromatic C–C signals at 132.26 ppm, C–O carbon signal at 70.92 ppm, and the epoxy (C–O–C) group carbons at 62.72 ppm that were observed in the GO spectrum disappeared in the RGO spectrum (see Figure 3).

Figure 2 also shows the spectrum of HDA-GS: 3160 cm^−1^ (O–H stretching), 2921 and 2852 cm^−1^ (aliphatic C–H stretching), 1375 cm^−1^ (aromatic C–N–C symmetric stretching), and 790 cm^−1^ (N–H out-of-plane stretching).

### 3.2. Wide-Angle X-Ray Diffraction (XRD)

The XRD results for the 2θ range of 2–32° for PVA hybrid films containing RGO and HDA-GS with various filler contents are shown in Figure 4 and Figure 5. In Figure 4, the intrinsic peak of GO appeared at 2θ = 9.61° (d = 9.19 Å), whereas no characteristic peak was observed in the acid-treated RGO. In the pure PVA films, the intrinsic characteristic peaks of PVA were found at 2θ = 19.7° (d = 4.50 Å); for the PVA hybrid films containing 0–10% *w*/*w* of RGO, the intensity of the PVA peaks was decreased, but no other peaks were observed [36,37].

In Figure 5, the intrinsic peak of HDA-GS was observed at 2θ = 2.65° (d = 33.29 Å), and the intrinsic characteristic peak of PVA was observed at 2θ = 19.7° (d = 4.50 Å), as in the case for RGO. For the HDA-GS/PVA hybrid, the characteristic peak of HDA-GS, 2θ = 2.65° (d = 33.29 Å) was observed in the same position but with a considerably lower intensity for the 3 to 10% *w*/*w* of HDA-GS hybrids; the peak at 19.7° remained almost the same for the 0 to 10% *w*/*w* hybrids, with only a slight decrease in intensity.

The results indicated that the dispersibility of both fillers in the PVA matrix polymer was good, regardless of the concentration of the fillers. However, in order to observe the aggregation of the filler and to investigate the dispersibility of graphene in more detail, it was necessary to perform further characterizations using an electron microscope.

### 3.3. Morphology

Using the results of the confirmed nanofillers, FE-SEM was used to investigate the fracture characteristics of pure PVA films and PVA hybrid films according to their graphene content. Figure 6 and Figure 7 show the SEM images of the hybrids containing different contents of RGO and HDA-GS. The pure PVA film in Figure 6 showed no distinct morphological features. However, in the PVA composite films containing 3–10% *w*/*w* RGO, most of the RGO demonstrated a uniform orientation and a flat plate shape. However, as the concentration of RGO increased from 5 to 10% *w*/*w*, the thickness and size of graphene gradually increased. The degree of graphene dispersion gradually decreased with increasing RGO concentration. However, most of the observed aggregates were smaller than 100 nm.

On the contrary, in the case of HDA-GS, even when the content of HDA-GS was increased to 10% *w*/*w*, no agglomeration of GS has been observed. However, compared to 3% *w*/*w* HDA-GS hybrid, we could observe a little bit of the graphene (see Figure 7). These results could be explained by the fact that, unlike RGO, HDA-GS contains many organic alkyl moieties and hydroxyl groups, and thus exhibited better mixing with the PVA matrix and showed a more uniform shape. The large number of hydroxy groups contained in HDA-GS enables hydrogen bonding with the PVA matrix, which had a strong influence on the thermal property, gas permeation, and electric conductivity of the hybrid described in the following sections.

### 3.4. Thermal Properties

Table 1 lists the DSC and TGA results of the PVA nanocomposite films containing the two nanofillers; their DSC behaviors are shown in Figure 8. The glass transition temperature (*T_g_*) of the pure PVA film was 67 °C. As the content of RGO or HDA-GS increased, the *T_g_* value of the hybrid film also increased. For example, at an RGO content of 5% *w*/*w*, the *T_g_* increased to 134 °C, but at 10% *w*/*w*, no glass transition was observed. In the case of HDA-GS, the *T_g_* value gradually increased from 67 to 73 °C as the HDA-GS content increased from 0 to 10% *w*/*w*. This can be explained by the fact that the polymer chain was sandwiched between rigid plate-like graphene layers, restricting its motion and eventually interfering with the chain-segmental motion of some polymer chains [38].

Similar to *T_g_*, the melting transition temperature (*T_m_*) value of the PVA hybrid film increased with increasing RGO and HDA-GS content in the PVA hybrid. A 13 °C increase to 174 °C was observed at 3% *w*/*w* RGO, but no *T_m_* was observed at 10% *w*/*w*. Similarly, the *T_m_* value of the HDA-GS hybrid film steadily increased from 161 to 173 °C as the HDA-GS content increased from 3 to 7% *w*/*w*. This indicated that the two graphene nanofillers played an important role in enhancing the thermal properties of PVA. Many researchers [30,37,38,39] have reported that the increase in *T_m_* is due to the heat shielding effect of the rigid and strong graphene fillers dispersed in the polymer chain, which greatly affects their melting behavior. The DSC thermograms of the PVA hybrids with various HDA-GS contents are shown in Figure 8. When the nanofiller was included, the peak was stronger than that of the pure PVA. As the amount of the filler was increased to 7% *w*/*w*, the intensity of the peak also increased. Hence, the HDA-GS appears to act as a nucleating agent [37,39].

However, for 10% *w*/*w* HDA-GS, the *T_m_* of the hybrid film decreased by 17 °C (156 °C). As discussed above, when an excessive amount of the filler was used, the filler was not dispersed in the matrix polymer, and instead agglomerated, thus further decreasing the thermal properties [30]. Finally, the results indicated that the thermal properties differed depending on the type, content, and dispersion of the filler used.

From the initial decomposition temperature (*T_D_^i^*) obtained from the TGA used to measure thermal stability, the *T_D_^i^* at 2% initial decomposition increased gradually as the two filler contents increased. These results indicate that the high heat resistance of graphene-based fillers contributes significantly to the thermal stability of the PVA hybrid films [40,41] (Table 1). In the case of RGO, the RGO increased steadily from 226 to 251 °C when the RGO content increased from 0 to 10% *w*/*w*, and from 226 to 252°C as the HDA-GS increased from 0 to 7% *w*/*w*. However, at an HDA-GS content of 10% *w*/*w*, it decreased to 238°C (see Table 1). This occurred because the excess organic parts of the substituted alkyl groups of HDA-GS could not withstand high temperatures and were easily decomposed. The TGA results of the hybrid films containing RGO and HDA-GS are shown in Figure 9.

The residual amount of the films at 600 °C (weight residue at 600 °C, wt_R_^600^) increased gradually as the graphene content was increased to 10% *w*/*w* in both hybrids (see Table 1, Figure 9). The residual amount of RGO increased from 6 to 29% when the filler content was increased to 10% *w*/*w*, while that of HDA-GS was 12% at 10% *w*/*w*, which was lower than that of RGO. This result can be explained by the presence of the alkyl groups in HDA-GS, which have weak thermal stability, as described above.

### 3.5. Gas Permeation

The effect of the polymer nanocomposite film on the gas permeation can be explained by the shape of the filler itself and the morphology of the filler dispersed in the polymer composite. That is, when a rigid plate-like nanoscale graphene with a large axial ratio is present, the gas molecules cannot pass through the resulting curved pathways, and the gas permeability eventually decreases [42,43]. In general, the gas permeability is significantly affected by the gas diffusion path. In particular, complex factors such as the solubility and diffusion rate of the gas to the polymer are involved. Additionally, both the interactions between the polymer chains, and the interaction between the polymer chain and the gas, contribute to the diffusion rate of the gas [11]. As a result, the permeabilities of hybrids composed of filler particles dispersed in a polymer matrix have been reported in various studies [44,45]. In particular, single-layer graphene has a plate-like shape with a large surface area of 2,630 m^2^/g and is known to have excellent gas barrier properties [46,47,48].

The oxygen transmission rate (O_2_TR) of the PVA hybrid film is shown in Table 2. In order to compare the permeation values more reliably, the film thicknesses of all samples were kept as uniform as possible in the range of approximately 20 to 24 μm. The pure PVA film without dispersed graphene showed an oxygen transmission value of 7.29 cc/m^2^/day. The oxygen transmission decreased to reach its lowest value as the filler content was increased up to its critical concentration; it then increased as the amount of filler was increased further. For example, in the case of RGO, the permeability was <10^−2^ cc/m^2^/day for a filler content of 3% *w*/*w*. As explained previously, the permeability value was most likely reduced because the gas molecules could not easily pass through the curved paths created by the rigid plate-like graphene present in the PVA matrix. However, when the amount of filler was increased to 10% *w*/*w*, the permeability increased sharply to 39.23 cc/m^2^/day. These results indicated that above the critical concentration, the RGO particles aggregated, resulting in pinholes in the hybrid film that increased permeation. In the case of HDA-GS, the permeation was 0.98 cc/m^2^/day at 5% *w*/*w*, but increased to 31.27 cc/m^2^/day at 10% *w*/*w*. This result can also be explained by the agglomeration of the filler at or above a critical concentration. These results are in agreement with the electron microscopy results shown in Figure 6 and Figure 7.

In conclusion, RGO and HDA-GS showed the best oxygen barrier properties at 3 and 5% *w*/*w*, respectively. In the hybrid films containing 10% *w*/*w* of either of the fillers, the excess filler aggregated, reducing the gas barrier properties.

### 3.6. Electric Conductivity

Generally, pure carbon fillers such as carbon black or carbon nanotubes, which are composed mainly of carbon, have excellent thermal conductivity (5000 W/mK) and electrical conductivity (6000 S/cm) [49,50]. F-GS has the advantage of good dispersibility in the matrix polymer due to its organic functional groups. However, the process of modifying graphene with organic groups through chemical treatment transforms the *sp*^2^ double bonds of graphene into *sp*^3^ single bonds. For this reason, the use of F-GS as a filler is expected to interfere with the movement of electrons, resulting in a decrease in the electrical conductivity of the hybrids compared to pure graphene or RGO hybrids.

Table 3 shows the electrical conductivities of PVA hybrid films with various RGO and HDA-GS contents. In order to accurately compare the results, the thickness of all the film samples was made as uniform as possible, at about 42 to 48 μm. The pure PVA film without a dispersed filler showed an electrical conductivity value of 10^−10^ S/cm. However, when the amount of RGO was increased to 10% *w*/*w*, an electrical conductivity of 1.1 × 10^−2^ S/cm was obtained, which represented a 10^8^ improvement in conductivity. When the RGO content was increased even further to 40% *w*/*w*, a conductivity of 1.1 S/cm was obtained. However, this RGO content was too high, so we did not investigate any other properties other than conductivity for the 20 and 40% *w*/*w* RGO samples.

In the case of the HDA-GS PVA hybrid films, the change in the electrical conductivity with the filler content followed the same trend as for RGO (Table 3). For example, increasing the HDA-GS from 3 to 10% *w*/*w* resulted in a significant increase in conductivity from 10^−10^ S/cm to 7.3 × 10^−3^ S/cm. However, overall, the conductivity of RGO was better than that of HDA-GS at the same filler content. This was because RGO is a simple plate-shaped laminated structure between layers, whereas, as described above in HDA-GS, the graphene surface is covered with the long alkyl groups of hexadecylamine, disturbing the percolation path [51].

## 4. Conclusions

Two graphene nanofillers, RGO and HDA-GS, were synthesized using carbon as a starting material, and PVA hybrid films were prepared using these synthetic fillers. The morphology of the films was investigated, and it was confirmed that the graphene layers were dispersed in the PVA matrix in a flat plate shape. As the filler content increased, some graphene aggregated above the critical concentration, but the fillers generally maintained their nanoscale size. The thermal properties *T_g_*, *T_m_*, *T_D_^i^*, and wt_R_^600^ increased as the filler contents increased for both fillers. In the case of HDA-GS, the physical properties of the filler were reduced due to the agglomeration of the filler above the critical concentration.

The oxygen permeability of the PVA hybrids decreased with increasing filler content but increased above the critical concentration. However, the electrical conductivity of the PVA hybrid films increased steadily in proportion to the filler content in the polymer matrix. Even polymers with low filler contents were found to exhibit superior properties compared to pure PVA. The addition of RGO was more effective than that of HDA-GS with regards to the thermal properties, oxygen barrier property, and electrical conductivity of the PVA hybrid films.

Through the obtained results, we showed a simple and effective method to synthesis PVA nanocomposites using the solution intercalation method. Improvements in the thermal property, gas barrier, and electrical conductivity of the PVA hybrids were observed. It is expected that the newly obtained PVA hybrids will be useful in various applications such as polymer electrolyte fuel cells, packaging films, drug delivery, and permeation membranes.

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
