# Peer review of "Comparison of Properties of PVA Nanocomposites Containing Reduced Graphene Oxide and Functionalized Graphene"

_polymers, 2019, doi:10.3390/polym11030450_

Round 1

Reviewer 1 Report

In this work Authors synthesize and study a variety of properties including thermal and electrical of composites of two graphitic materials with PVA: RGO and a functionalized  hexadecylamine-graphene. Authors find that increase in graphitic gfillers enhances the conductivity and thermal properties however there is an optimal amount of a filler for gas permeation after which asuch permeation gets affected by the aggregation of the filler. Electrical properties of RGO-based films appear to be superior to those of hexadecylamine-graphene-based ones.

Here are few comments:

There is a plethora of scientific papers discussing a variety of properties of graphene or GO or RGO/PVA composites.The novelty of this paper is that it compares all the applicable RGO-PVA and HAD-GS composites. The introduction should clearly indicate how the current paper is different from the aforementioned multiple works.

Other comments include:

 hexadecylamine-graphene described here is actually  hexadecylamine-graphene oxide as many of the oxygen-containing functional groups remain; that may need to be addressed throughout the paper.

Line 44 "realize" may need to be changed to utilize

synthesis of PVA and RGO needs to be described in methods

Line 54 sentence structure needs to be revised

Line 156 sentence does not appear to be comprehensive and needs to be rewritten: also a suggestion to further use oxygen addends or moieties rather than oxygen molecules as those groups are bound

Line 168 The XRD results only indicate the retainment of PVA peak as HAD-GS has no features. It is then not an indication of excellent dispersibility, thus a milder wording needs to be used.

Line 202 RGO is a material not a phenomenon

Figures 7&8, as well as 9&10 need to be combined

Author Response

Dear Editor

This is my response to your comments regarding our paperComparison of Properties of PVA Nanocomposites Containing Reduced Graphene Oxide and Functionalized Graphene”

(polymers-455780) in polymers.

Thank you very much for the referee's comments. I have carefully revised the manuscript following the comments of the referee.

Response to Reviewer-1 comments

Point-1: The introduction should clearly indicate how the current paper is different from the aforementioned multiple works.

Response-1: As the referee pointed out, “In general, F-GS has its advantages - the epoxy groups via deoxygenation [30].” was added in Introduction. 

Point-2: For hexadecylamine-graphenes containing oxygen atoms.

Response-2: “The synthesized HAD-GS still contains oxygen elements” was added in Section 2.4 Synthesis of HAD-GS.

Point-3: “realize” may be need to be changed to “utilize”

Response-3: As pointed out, we changed realize to utilize in line 46.

Point-4: Synthesis of PVA/RGO nanocomposite.

Response-4: The synthetic method of the PVA/RGO hybrid followed the same method as the PVA/HDA-GS hybrid. The synthetic method of the HDA-GS hybrid describe in Section 2.5. “HDA-GS was dispersed in DMAc - in a 40 °C oven over 72 h to obtain a film.”

Point-5: Various chemical structures of functionalized-graphenes 

Response-5: Figure 1 shows the chemical structures of various F-GSs that were synthesized for polymer nanocomposites. See Figure 1.

Point-6: Sentence needs to be rewritten: The oxygen moiety is better than the oxygen molecule.

Response-6: As pointed out, we changed molecules to moieties in line 185.

Point-7: A milder wording needs to be used.

Response-7: As pointed out, we changed excellent to good in line 213.

Point-8: RGO is a material not a phenomenon

Response-8: As reviewer pointed out, “no agglomeration of GS has been observed.” has been modified in line 222. 

Point-9: Figures 7 and 8, as well as 9 and 10 need to be combined.

Response-9: As reviewer pointed out, Figures 7 & 8, as well as 9 & 10 have been combined. See Figures 8 and 9. 

I hope this revision is satisfactory for your further process. 

Best,

Jin-Hae Chang

Professor

Reviewer 2 Report

Work titled: “Comparison of properties of PVA nanocomposites containing reduced graphene oxide and functionalized graphene” described fabrication and characterization steps of prepared composites based on functionalized and oxidized graphene flakes. This work is interesting however some critical motivation and potential applications are missing. For example, fabricated graphene flakes in both batches were not analyzed in respect to their sizes. Such data can be helpful for comparison to other published composite materials. Some additional remarks are listed below.

Line 21. Industrial applications are not listed in this study. There is no evidence of prepared devices.

Line 50. There are recent studies on the nanostructuted GO and polymers composites such as: NATURE COMMUNICATIONS | 6:8817 | DOI: 10.1038/ncomms9817, Angew.Chem.Int.Ed.2016,55,12516–12521, and Adv.Mater.2016, 28, 8365–8370)

Line 182. Can the author quantitatively compare how does the spacing between 2d materials changes after adding more PVA polymers?

Line 199. Figure capture is not detailed. Preferably, additional thickness measurements of the composites and formed particles will give better judgment of difference between particles. Statistical histogram can be helpful to see the difference.   

Line 207. The list of various properties can be listed here.

Line 277. The method of polymer thickness measurements is not described.

Line 321. Conclusion part missing comparison to other published results with the better stress of novel properties.

Author Response

Dear Editor

This is my response to your comments regarding our paperComparison of Properties of PVA Nanocomposites Containing Reduced Graphene Oxide and Functionalized Graphene”

(polymers-455780) in polymers.

Thank you very much for the referee's comments. I have carefully revised the manuscript following the comments of the referee.

Response to Reviewer-2 comments

Point-1: Some critical motivation and potential applications are missing.

Response-1: As the reviewer pointed out, “In general, F-GS has its advantages - ultrathin films useful as transparent devices and electrodes.” was added in Introduction. 

Point-2: Industrial applications are not list. 

Response-2: “Graphite-based materials are commercial - ultrathin films useful as transparent devices and electrodes.” was added in Introduction. 

Point-3: Supplement reference.

Response-3: As pointed out, we added reference in reference section. See reference No. 25.

Point-4: Can the author quantitatively compare how does the spacing between 2d materials changes after adding more PVA polymers?

Response-4: Unfortunately, we could not test the XRD. Especially, in case of HDA-GS, since the amount of synthesis was too small (1 g), various experiments were not performed. We are so sorry for this.

Point-5: Statistical histogram of the fillers can be helpful to see the difference.   

Response-5: Since the filler used in the polymer composite is in a state of being mixed or clump in the matrix polymer, accurate thickness of the filler can not be measured.

Point-6: Describe the type of concrete properties

Response-6: See line 227. “the thermal property, gas permeation, and electric conductivity of the hybrid described in the following sections.” was added.

Point-7: The method of polymer film thickness measurements.

Response-7: “The thickness of the obtained film was measured using a digital micrometer (Mituyoto 293-240).” was added in Section 2.6. Characterizations.

Point-8: Comparison to other published results with the better stress of novel properties in Conclusion.

Response-8:Through the obtained results, we showed - packaging films, drug delivery, and permeation membranes.” was added in Conclusion section. 

I hope this revision is satisfactory for your further process. 

Best,

Jin-Hae Chang

Professor
